# *Ex-vivo* cultured human corneoscleral segment model to study the effects of glaucoma factors on trabecular meshwork

**Ramesh B. Kasetti**❦, **Pinkal D. Patel**❦, **Prabhavathi Maddineni, Gulab S. Zode**❦*

Department of Pharmacology and Neuroscience and the North Texas Eye Research Institute, University of North Texas Health Science Center at Fort Worth, Fort Worth, TX, United States of America

❦ These authors contributed equally to this work.
* gulab.zode@unthsc.edu

**Data Availability Statement:** All relevant data are within the manuscript and its Supporting Information files.

## Abstract

Glaucoma is the second leading cause of irreversible blindness worldwide. Primary open angle glaucoma (POAG), the most common form of glaucoma, is often associated with elevation of intraocular pressure (IOP) due to the dysfunction of trabecular meshwork (TM) tissues. Currently, an *ex vivo* human anterior segment perfusion cultured system is widely used to study the effects of glaucoma factors and disease modifying drugs on physiological parameters like aqueous humor (AH) dynamics and IOP homeostasis. This system requires the use of freshly enucleated intact human eyes, which are sparsely available at very high cost. In this study, we explored the feasibility of using human donor corneoscleral segments for modeling morphological and biochemical changes associated with POAG. Among the number of corneas donated each year, many are deemed ineligible for transplantation due to stringent acceptance criteria. These ineligible corneoscleral segments were obtained from the Lions Eye Bank, Tampa, Florida. Each human donor anterior corneoscleral segment was dissected into four equal quadrants and cultured for 7 days by treating with the glaucoma factors dexamethasone (Dex) or recombinant transforming growth factor (TGF) β2 or transduced with lentiviral expression vectors containing wild type (WT) and mutant myocilin. Hematoxylin and Eosin (*H&E*) staining analysis revealed that the TM structural integrity is maintained after 7 days in culture. Increased TUNEL positive TM cells were observed in corneoscleral quadrants treated with glaucoma factors compared to their respective controls. However, these TUNEL positive cells were mainly confined to the scleral region adjacent to the TM. Treatment of corneoscleral quadrants with Dex or TGFβ2 resulted in glaucomatous changes at the TM, which included increased extracellular matrix (ECM) proteins and induction of endoplasmic reticulum (ER) stress. Western blot analysis of the conditioned medium showed an increase in ECM (fibronectin and collagen IV) levels in Dex- or TGFβ2-treated samples compared to control. Lentiviral transduction of quadrants resulted in expression of WT and mutant myocilin in TM tissues. Western blot analysis of conditioned medium revealed decreased secretion of mutant myocilin compared to WT myocilin. Moreover, increased ECM deposition and ER stress induction was observed in the TM of mutant myocilin transduced quadrants. Our findings suggest that the *ex-vivo* cultured

**Funding:** These studies were supported by the National Institutes of Health; EY028616 (GSZ) and EY026177 (GSZ).

**Competing interests:** The authors have declared that no competing interests exist.

human corneoscleral segment model is cost-effective and can be used as a pre-screening tool to study the effects of glaucoma factors and anti-glaucoma therapeutics on the TM.

## Introduction

Glaucoma is the second leading cause of irreversible blindness affecting nearly 70 million people worldwide [1]. Primary open angle glaucoma (POAG) is the most common form of glaucoma. The majority of POAG patients experience chronic elevation in intraocular pressure (IOP), which is the only modifiable and treatable risk factor associated with the disease [2]. The balance between aqueous humor secretion from the ciliary body and its outflow through the trabecular meshwork (TM) is critical for maintenance of IOP homeostasis within an acceptable physiological range. In POAG, there is an increased resistance to the aqueous humor outflow through the TM, which leads to IOP elevation [3, 4]. However, the molecular pathology underlying TM dysfunction and IOP elevation are yet to be fully understood. Moreover, the majority of current treatments do not address the underlying pathology of glaucomatous TM damage. Of note, Rho-kinase inhibitors are the newest class of glaucoma drugs that directly target the TM to increase aqueous outflow [5–7]. Several factors are involved in glaucomatous TM damage, resulting in elevation of IOP. *MYOC* was the first glaucoma gene identified [8] and is responsible for approximately 4% of POAG [9]. Although the normal role for wild type (WT) myocilin is unknown, mutations in *MYOC* cause a deleterious gain-of-function leading to inhibition of its secretion [10], intracellular accumulation and protein stress within TM cells [11–13]. Another major contributor to glaucomatous TM damage is the increased expression of the pro-fibrotic cytokine, transforming growth factor (TGF) β2 in the aqueous humor and TM of POAG patients [14–17]. In addition, glucocorticoid (GC) therapy can cause ocular hypertension (OHT) and development of iatrogenic open angle glaucoma in susceptible individuals [18, 19]. A potent GC, dexamethasone (Dex) is known to cause pathological alterations in the TM such as increased ECM synthesis and deposition, induction of ER stress, reduced TM phagocytic function and cytoskeletal remodeling [20–28]. Since clinical presentations of GC-induced glaucoma are similar to POAG in many ways [29], several laboratories have utilized GC-induced OHT to understand glaucomatous TM dysfunction. Treatment with Dex or TGFβ2 elevates IOP in mice *in vivo* [30–32] and in perfusion cultured human anterior segments *ex vivo* [33–35]. Both Dex and TGFβ2-induced OHT are associated with changes to the TM cytoskeleton [33, 36] and ECM deposition [28], which stiffen the TM [35, 37].

There is an unmet need in glaucoma research for a versatile model system that closely recapitulates the disease condition in humans. Although there are remarkable anatomical and physiological similarities between humans and animal models of glaucoma, the results obtained with animal models rarely translate to disease modifying outcomes in human patients. The opportunity to work with gifted human donor tissues is a unique privilege shared by the ocular research community. Perfusion cultured human anterior segments have been frequently used as an *ex-vivo* model in glaucoma research [34, 35, 38–41]. The anterior segments used in perfusion culture are obtained from freshly enucleated human eyes post-mortem for studying physiological parameters like AH dynamics and IOP homeostasis. However, sparse availability, very high cost and high failure rates have impeded the extensive use of the perfusion culture model in glaucoma research.

The cornea is the most commonly donated organ of the human body worldwide. In the United States, 116,990 corneas were donated in 2016, of which only 63,596 were used for transplantation [42]. Due to stringent transplantation eligibility criteria, the majority of corneas are deemed ineligible [43]. These ineligible corneoscleral segments are easily available with lower processing cost compared to intact whole globes. These corneoscleral segments consist of a complete cornea, sclera, and an intact TM rim. Moreover, the patient medical history is readily available. Several labs utilize these tissues to isolate primary TM cells. Since these corneoscleral segments can be cultured in stationary media for several days, it is possible to utilize corneoscleral segments to test the effects of glaucoma factors on intact TM tissues. This study investigates the feasibility of using these corneoscleral segments to detect the morphological and biochemical changes associated with glaucoma pathogenesis. To test this corneoscleral segment model in recapitulating the disease phenotype, we treated these explants with factors known to induce glaucomatous pathology including Dex, TGFβ2, and the mutant form of myocilin protein (Y437H). We examined whether these glaucoma factors induce similar pathological changes in the TM of *ex-vivo* cultured human corneoscleral segments.

## Material and methods

### Antibodies

Antibodies were purchased from the following sources; fibronectin (catalog# ab2413, Abcam), Collagen type IV (catalog # SAB4500369, Sigma-Aldrich), ATF4 (catalog# sc-200, Santa Cruz Biotechnology), CHOP (catalog# 13172, Novus Biologicals), GAPDH (catalog# 3683; Cell Signaling Technology), GRP78 (N-20, catalog# sc-1050, Santa Cruz Biotechnology), FLAG (catalog# A8592, Sigma), Myocilin (36–135, catalog# H00004653, Abnova).

### *Ex-vivo* culture of human corneoscleral segments

Human corneoscleral segments with intact TM rim were obtained from the Lions Eye Bank (Tampa, Florida) in accordance with Declaration of Helsinki guidelines. The corneoscleral segments were dissected into 4 equal quadrants and each quadrant was cultured in a 24-well plate using DMEM medium supplemented with 10% FBS, L-glutamine and 1% Pen-strep (**Fig 1**). Cultured medium was changed every two days. Quadrants were treated either with 100nM Dex or 5 ng/ml recombinant TGFβ2 (in 0.5% FBS containing medium) or transduced with

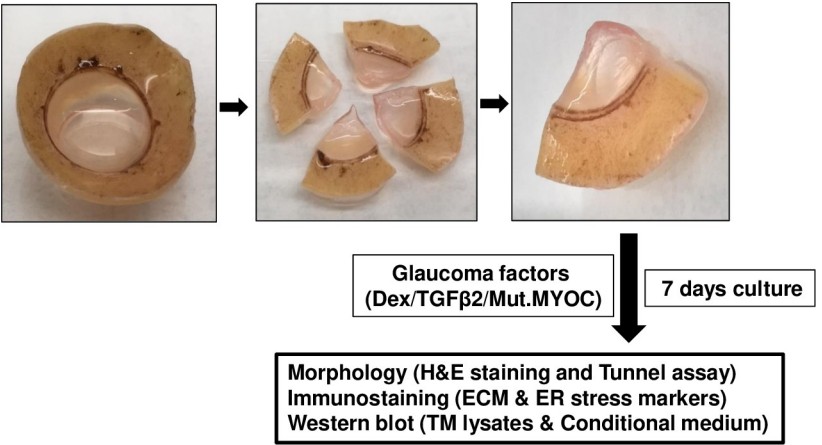

**Glaucoma factors (Dex/TGFβ2/Mut.MYOC)**

**7 days culture**

**Morphology (H&E staining and Tunnel assay)**
**Immunostaining (ECM & ER stress markers)**
**Western blot (TM lysates & Conditional medium)**

**Fig 1. Schematic showing the experimental design.** Human corneoscleral segments with an intact TM rim were dissected into 4 equal quadrants and each quadrant was cultured and treated with glaucoma-inducing factors for 7 days. Each quadrant was then either fixed, or TM tissue and conditioned medium was isolated. Fixed tissues were utilized for H&E and immunohistochemical analysis while TM tissue and conditioned medium was analyzed by Western blot.

lentiviral vectors expressing FTS (FLAG and S) tagged WT and mutant myocilin for 7 days with appropriate controls.

## H & E staining

Cultured corneoscleral quadrants were fixed overnight in freshly prepared 4% paraformaldehyde in PBS. Fixed tissues were washed 3 times with PBS, dehydrated with ethanol and embedded in paraffin. Paraffin-embedded quadrants were then sectioned at 5 µm thickness, deparaffinized in xylene and rehydrated and stained with hematoxylin and eosin.

## TUNEL assay

TUNEL assay was performed on human cultured corneoscleral quadrant sections using a TUNEL Apo-Green Detection Kit (Biotool, Houston, TX, USA) in accordance with the manufacturer's recommended protocol. The quadrants were cultured at zero or 7 days either with 10% FBS or 0.5% FBS containing DMEM culture medium. Quadrants were then fixed in 4% PFA, processed, and embedded in paraffin. Five-micron sections were cut using a microtome. Sections were dewaxed, rehydrated, and permeabilized by incubating with 20µg/ml proteinase K solution for 20 minutes at room temperature. Sections were then incubated with equilibration buffer for 10 minutes and allowed 60 minutes incubation with the TUNEL reaction mixture (containing Apogreen labeling mix and Recombinant TdT enzyme) at 37˚C in the dark. Following 3 washes in PBS, sections were mounted using DAPI mounting solution. TUNEL-positive green fluorescence images were taken using a Keyence microscope (Itasca, IL, USA).

## Immunostaining

Cultured corneoscleral segments were fixed overnight in freshly prepared 4% PFA at 4˚C, processed and embedded in paraffin. The paraffin embedded quadrants were cut into 5 µm sections. Sections were dewaxed, rehydrated and subjected to antigen retrieval by incubating sections in citrate buffer (pH 6) for 2hrs in a water bath at 60˚C. Sections were then allowed to cool to room temperature and blocked with 0.2% Triton-X-100 containing 10% normal goat serum for 2hrs. Slides were incubated overnight with primary antibody (1:500 dilution) in blocking buffer, and then washed 3 times with PBS followed by a 2-hour incubation with the appropriate Alexa Fluor secondary antibodies (1:1000; Life technologies, Grand Island, NY, USA). Sections were mounted with DAPI-mounting solution. Images were captured using a Keyence microscope (Itasca, IL, USA).

## Western blot analysis

TM tissue from each cultured corneoscleral quadrant was peeled off and lysed in radioimmunoprecipitation (RIPA) lysis buffer containing protease inhibitors (Life technologies, Grand Island, NY, USA). The insoluble material was separated out by centrifugation and supernatants (lysates) were collected. The protein concentration in the lysates was measured using the DC protein assay kit (Bio-Rad, CA, USA). Equal amounts of tissue lysates were loaded on denaturing 4–12% gradient polyacrylamide ready-made gels (NuPAGE Bis-Tris gels, Life technologies, Grand Island, NY, USA) and transferred onto PVDF membranes. Blots were blocked with 10% non-fat dried milk for 1 hour then incubated overnight with specific primary antibodies at 4˚C on a rotating shaker. The membranes were washed three times with PBST and incubated with corresponding HRP-conjugated secondary antibody for 90 minutes. The proteins were then visualized using ECL detection reagents (SuperSignal West Femto Maximum Sensitivity Substrate; Life Technologies, Grand Island, NY, USA). To analyze secreted proteins,

conditioned medium (1ml) was mixed with 10 μl resin (StrataClean Resin; Agilent, Inc., Santa Clara, CA, USA) and rotated overnight at 4˚C to concentrate protein from the medium. Samples were centrifuged and the pellet was analyzed by Western blot as described above. Coomassie staining of the gel was performed as a loading control to ensure equal protein loading.

## Lentivirus preparation

HEK293 cells were co-transfected with lentiviral construct containing FTS (FLAG and S) tagged wild type or mutant myocilin and lentivirus packing plasmids (p9.81 and pMDG) using lipofectamine 3000. Six hours post-transfection, medium containing lipid-DNA complexes was replaced with 10ml of lentivirus packing medium (DMEM with 5% FBS). The first collection of lentivirus supernatant was carried out after 24hrs of transfection and the second collection after 52hrs of transfection. The first and second collections were pooled and centrifuged to remove any cell debris and filtered through a 45μm filter. Supernatants containing virus particles were aliquoted and stored at -80˚C. Different concentrations of lentivirus supernatants containing WT or mutant myocilin were tested for transduction efficiency on TM tissue of cultured corneoscleral segments. One ml of supernatant per well of a 24 well plate was determined as an effective titer.

## Statistical analysis

Statistical analyses were performed using GraphPad Prism (GraphPad Software, La Jolla, CA). All data are represented as mean ± SEM. The data between two groups were analyzed using the unpaired two-tailed student's t-test. TUNEL-positive cells in the TM region were counted and analyzed using one-way analysis of variance, followed by Tukey's multiple comparison test. *P* less than or equal to 0.05 was considered significant.

# Results

## TM structural integrity and cellularity is maintained in the cultured human corneoscleral segments

Human corneoscleral segments with an intact TM rim were received from the eye bank and dissected into 4 equal quadrants. Each quadrant was cultured in a 24-well plate and treated with glaucoma-inducing factors for 7 days. Each quadrant was then either fixed, or TM tissue and conditioned medium was isolated. Fixed tissues were utilized for *H & E* and immunohistochemical analysis while TM tissues and conditioned medium were analyzed by Western blot (**Fig 1**).

Preservation of tissue structural integrity and cell viability is crucial for developing an *ex-vivo* tissue culture model. Using H&E staining and the TUNEL assay (**Fig 2**), we first examined whether TM structural integrity and cellularity is maintained after 7 days in an *ex-vivo* culture conditions. We cultured the corneoscleral segments with two different serum conditions; i) Standard serum media (10% FBS, suitable for Dex and myocilin treatments) and ii) Low serum media (0.5% FBS, suitable for TGFβ2 treatment). *H&E* analysis revealed no significant changes in the TM structural morphology among the tested groups (**Fig 2**, upper panel). A small number of TUNEL-positive cells were observed in the TM at zero-day cultured corneoscleral segments. The numbers of TUNEL-positive TM cells were identical between the zero-day and 7-day cultured quadrants with standard serum conditions. On the contrary, a significant increase in TUNEL-positive cells were observed in the TM region with low serum conditions. The numbers of TUNEL-positive cells were higher in the scleral tissue compared to the TM region (**Fig 2**, lower panel).

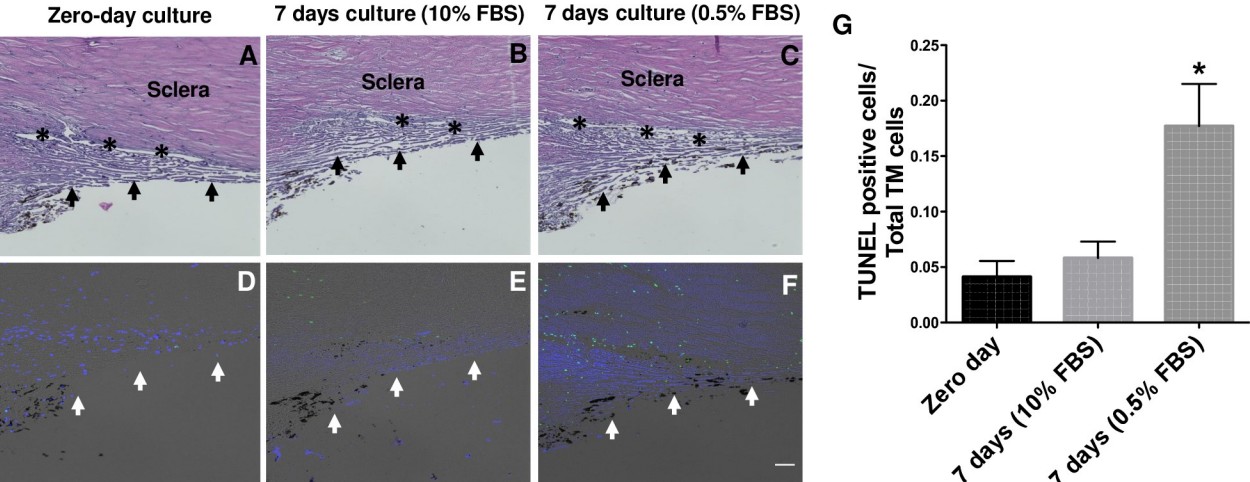

**Fig 2. TM structural integrity is maintained in an *ex-vivo* cultured corneoscleral segment.** H&E staining (upper panel) and TUNEL assay (lower panel) was performed in human corneoscleral quadrants cultured at zero days (**A&D**), 7 days with 10% FBS (**B&E**) and 7 days with 0.5% FBS (**C&F**) containing medium. H&E staining revealed no obvious morphological changes after 7-days of culture compared to zero-days of culture. The total number of TUNEL positive cells over TM cells (DAPI positive) in TM region were shown graphically (**G**). No difference in TUNEL-positive TM cells was observed between the zero- day and 7-day cultured group with 10% FBS. A significant increase in the TUNEL- positive cells was observed in quadrants cultured with 0.5% FBS. (n = 4 biological replicates per group, one-way ANOVA, *$p < 0.05$), scale bar is 50µm. Arrows indicate the TM region; asterisks represent the Schlemm's canal.

## Dex- or TGFβ2-treatment increases TM cell apoptosis in an *ex-vivo* cultured human corneoscleral segments

Progressive loss of TM cells is associated with POAG [44, 45]. We next determined whether Dex- or TGFβ2-treatment leads to accelerated TM cell death in cultured human corneoscleral segments. Cultured human corneoscleral segments were treated with Dex or TGFβ2 along with appropriate controls. H&E staining revealed normal iridocorneal angle tissues including intact TM and SC. Interestingly, we observed significantly increased TUNEL-positive TM cells

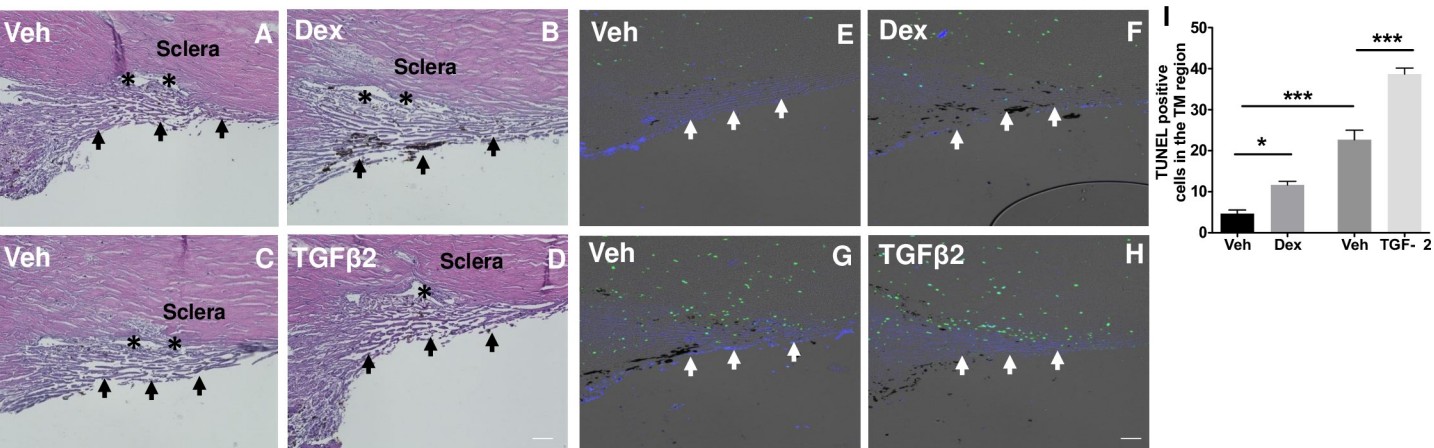

**Fig 3. Dex- or TGFβ2-treatment leads to increased apoptosis of TM cells.** H&E staining (left panel) and TUNEL assay (right panel) of cultured corneoscleral segments treated with either Dex (100nM) (**B&F**) or TGFβ2 (5ng/ml) (**D&H**) is compared with corresponding quadrants treated with vehicle (**A&E** or **C&G**) for 7-days. The total number of TUNEL-positive cells in the TM region were shown graphically (**I**). Increased TUNEL-positive (green) cells at the TM region was observed in both Dex- and TGFβ2-treated groups. (n = 3 biological replicates per group), scale bar is 50µm. Arrows indicate the TM region; asterisks represent the Schlemm's canal. N = 3 each; one-way ANOVA; *$p < 0.05$, ***$p < 0.0001$.

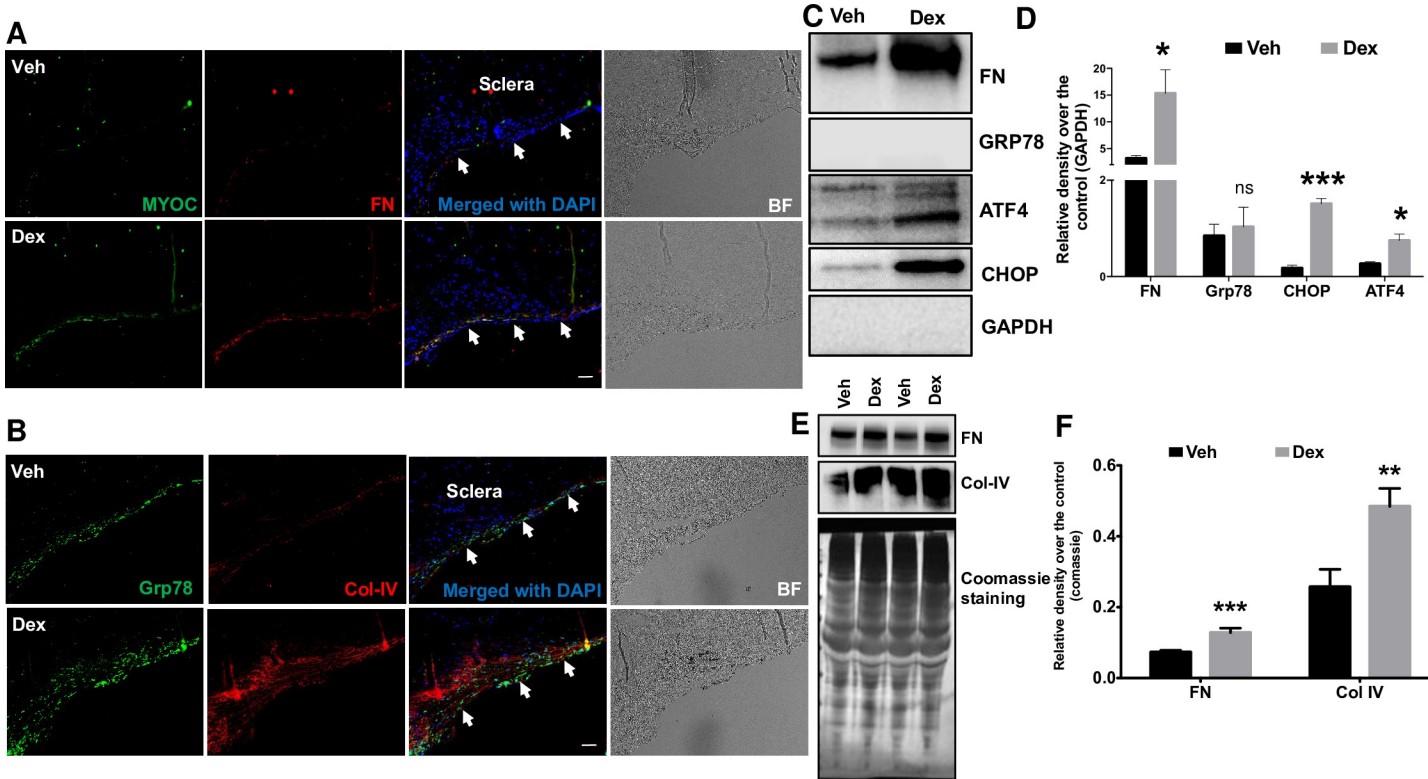

**Fig 4. Increased ECM accumulation and ER stress induction in the TM of Dex-treated cultured corneoscleral segments. A)** Immunostaining for myocilin and fibronectin, **B)** GRP78 (ER stress marker) and collagen IV in quadrants cultured for 7 days treated with the vehicle (0.1% ethanol) and Dex (100nM). Dex treatment prominently increased myocilin, fibronectin, collagen IV and GRP78 staining in the TM region. (n = 4 biological replicates, scale bar is 50μm). Western blot and densitometric analysis for FN (ECM marker) and ATF4, CHOP and GRP78 (ER stress markers) in TM tissue lysates (**C-D**) of vehicle- and Dex- (100nM) treated cultured corneoscleral segments. Dex treatment led to a significant increase in the ECM marker, FN (n = 3 biological replicates) and ER stress markers CHOP and ATF4 but not GRP78 (n = 4 biological replicates). Note that densitometric analysis included only Dex responders. Similarly, conditioned medium (**E-F**) from Dex-treated corneoscleral segments showed a significant increase in ECM proteins FN and Col IV (n = 8); unpaired t-test, $^{*}P<0.05$, $^{**}P<0.01$, $^{***}P<0.001$). Arrows indicate the TM region.

in human corneoscleral quadrants treated with Dex or TGFβ2 compared to their respective controls (**Fig 3**).

## Dex treatment increases ECM deposition and ER stress markers in the TM of cultured corneoscleral segments

Our previous studies demonstrated that increased ECM deposition and induction of ER stress is associated with human glaucomatous TM tissues and in the TM of mouse models of Dex-induced ocular hypertension [8, 30, 31, 46]. Moreover, Dex is known to increase myocilin expression [8, 47]. Here, we examined whether similar changes can be induced by Dex treatment in the TM of cultured human corneoscleral segments. Immunohistochemical analysis revealed increased expression of myocilin, fibronectin, collagen IV (ECM markers) and GRP78 (ER stress marker) in Dex-treated corneoscleral quadrants compared to the vehicle-treated controls (**Fig 4A and 4B**). Western blot and densitometric analysis for ECM and ER stress markers in TM tissue lysates and conditioned medium also revealed that Dex treatment increased ECM (fibronectin and Col-IV) and ER stress (ATF4, CHOP and GRP78) proteins (**Fig 4C–4F**). However, despite the increasing trend, GRP78 expression in the Dex-treated TM was not significant when compared to control. We next examined whether TM cellularity is a

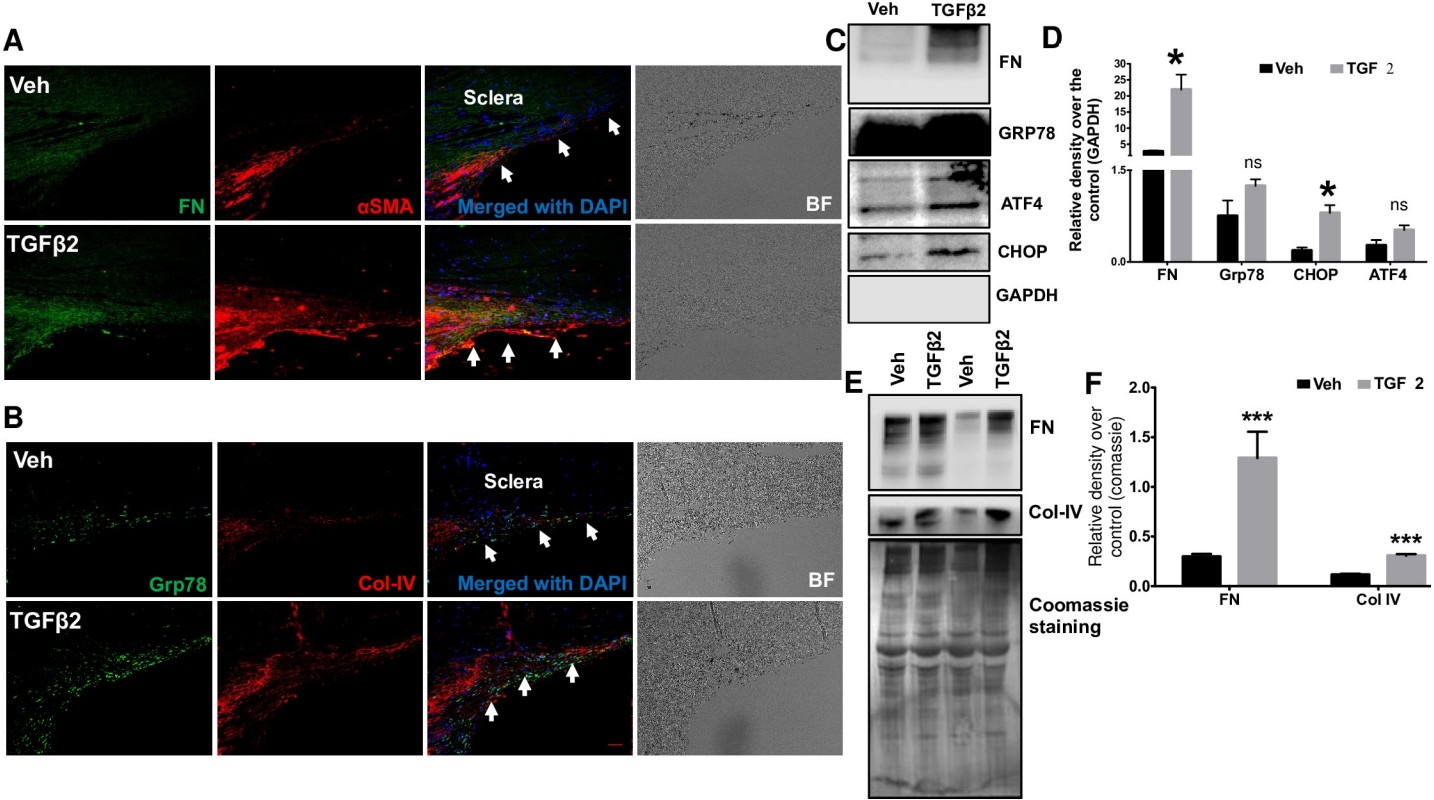

**Fig 5. Increased ECM accumulation in the TM of TGFβ2-treated cultured corneoscleral segments.** (**A** and **B**) Immunostaining for fibronectin (FN), collagen IV (Col-IV), αSMA and GRP78 in vehicle and TGFβ2 (5ng/ml) treated cultured corneoscleral segments. (n = 4 biological replicates, scale bar is 50μm). Western blot and densitometric analysis for FN, Col-IV (ECM markers), ATF4, CHOP, GRP78 in the TM tissue lysates (**C-D**) and conditioned medium (**E-F**) of vehicle and TGFβ2-treated cultured corneoscleral quadrants (n = 4 biological replicates for lysates and n = 8 for the conditioned medium), unpaired t-test, *$P<0.05$, **$P<0.01$, ***$P<0.001$. Arrows indicate the TM region.

contributing factor in Dex-induced changes in the TM of cultured corneoscleral segments. The number of DAPI-stained cells in the TM region were counted and no significant difference between vehicle (121±13.7) and Dex (133.8±8.1) treated groups (n = 5, two-tailed student's *t*-test) was found.

## TGFβ2 treatment induces ECM deposition in the TM of *ex-vivo* cultured human corneoscleral segments

TGFβ2 is known to alter production, degradation and composition of ECM in TM cells [35, 48–51]. Several studies have reported increased ECM deposition in human TM cells treated with TGFβ2 [35, 50, 51]. Increased fibronectin levels were observed in *ex-vivo* human anterior segment perfusates treated with TGFβ2 [35]. Here, we examined whether TGFβ2 treatment also increased ECM deposition in the TM of cultured human corneoscleral segments by analyzing fibronectin and collagen-IV levels, the major ECM components of the TM. Immunohistochemical analysis indicated an increase in fibronectin, collagen-IV (ECM proteins) and GRP78 (ER stress marker) staining in the TM of cultured corneoscleral quadrants treated with TGFβ2 compared to vehicle-treated controls (**Fig 5A and 5B**). Consistent with previous reports [35, 52], increased α-smooth muscle actin (αSMA) staining was observed in the TM of the TGFβ2-treated group (**Fig 5A**). Western blot and densitometric analysis on TM tissue lysates (**Fig 5C and 5D**) obtained from cultured corneoscleral segments revealed a significant

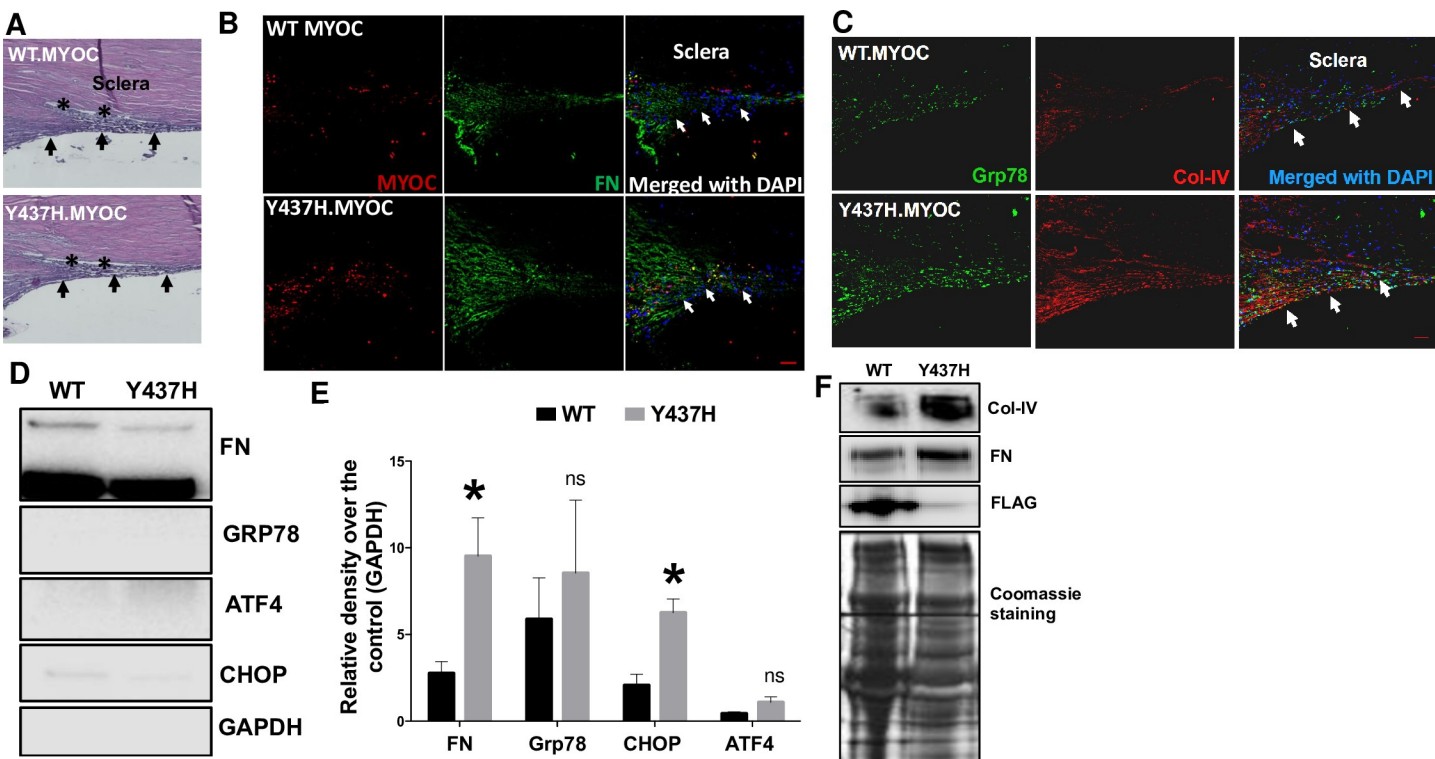

**Fig 6. Lentiviral expression of mutant myocilin induces ER stress and ECM changes in the TM of cultured corneoscleral segments.** The cultured corneoscleral quadrants were transduced with FTS tagged (FLAG & S tag) WT myocilin or mutant myocilin (Y437H) expressing lentiviral particles (1ml of lentivirus supernatant) for 7 days. **A**) H&E staining and (**B&C**) immunostaining for myocilin, FN, GRP78 and Col-IV in cultured corneoscleral segments transduced with WT or mutant myocilin. Increased myocilin staining was observed in the TM of mutant myocilin-transduced quadrants compared to WT myocilin. In addition, increased FN and Col-IV staining indicate more ECM accumulation in the TM of mutant myocilin-transduced quadrants (n = 3 biological replicates, scale bar is 50μm). Western blot and densitometric analysis of TM tissue lysates (**D-E**) and conditioned medium (**F**) obtained from cultured quadrants transduced with WT and mutant myocilin lentiviral expression vectors. A significant increase in the ECM marker FN (n = 6 biological replicates) and the ER stress marker CHOP (n = 3 biological replicates) was observed in mutant myocilin-transduced TM tissue lysates. Similarly, conditioned medium (**F**) from mutant myocilin-treated corneoscleral segments showed increases in ECM proteins FN and Col IV. Moreover, WT myocilin was detected in conditioned media of WT myocilin-transduced quadrants while no myocilin was detected in quadrants expressing mutant myocilin indicating that expression of mutant myocilin inhibits its secretion and accumulates in the TM cells. Unpaired t-test, *$P<0.05$, **$P<0.01$, ***$P<0.001$. Arrows indicate the TM region.

increase in the ECM marker, fibronectin and ER stress marker, CHOP. Although ER stress markers GRP78 and ATF4 showed an increasing trend in expression compared to control, the difference was not significant. Furthermore, conditioned medium (**Fig 5E and 5F**) of cultured corneoscleral quadrants treated with TGFβ2 revealed a significant increase in ECM proteins fibronectin and collagen-IV (ECM proteins). The number of DAPI stained cells in the TM region were not altered significantly between vehicle (119.8±10.5) and TGFβ2 (126.2±10.1) treated samples (n = 5, two-tailed student's *t*-test).

## Lentivirus-mediated expression of mutant myocilin increases ECM deposition and ER stress induction in the TM

We have previously reported that expression of the mutant form of human myocilin (Y437H) in human TM cells and in the TM of transgenic mice resulted in excessive ECM deposition and ER stress induction [11, 13, 53]. Therefore, we examined whether lentivirus-mediated expression of mutant myocilin protein (Y437H) induces ER stress and ECM deposition in the TM of cultured corneoscleral segments. Lentivirus-mediated transduction of corneoscleral segments showed no adverse effects on TM morphology as determined by H&E analysis (**Fig 6A**).

Transduction of corneoscleral quadrants with WT and mutant myocilin-expressing lentiviral particles resulted in expression of myocilin protein in the TM as determined by immunohistochemistry. However, the inability of the TM cells to secrete the mutant form of myocilin resulted in increased staining of myocilin in the TM of mutant myocilin transduced quadrants compared to WT myocilin-transduced controls (**Fig 6B**). This finding is consistent with our previous observations and indicates that mutant myocilin is secretory incompetent and accumulates inside the TM cells [11, 13, 53]. Immunohistochemical analysis showed an increase in FN, Col-IV and GRP78 in the TM of corneoscleral quadrants transduced with mutant myocilin compared to WT controls (**Fig 6B and 6C**). Western blot and densitometric analysis on TM tissue lysates showed a significant increase in the ECM marker FN and the ER stress marker CHOP in mutant myocilin-transduced quadrants compared to WT control (**Fig 6D and 6E**). Although the difference was not significant, we also observed an increasing trend in the expression of GRP78 and ATF4 in TM tissues of mutant myocilin-transduced quadrants. Western blot analysis of conditioned medium revealed an increase in fibronectin and Col-IV protein levels in the mutant myocilin-transduced corneoscleral segments (**Fig 6F**). No significant difference in the number of DAPI stained cells in the TM region was observed between WT (139.2 ±8.1) and mutant myocilin (119±10.1) transduced quadrants (n = 5, two-tailed unpaired student's *t*-test).

## Discussion

An *ex vivo* human anterior segment perfusion culture model has been used to study the effects of drugs on glaucomatous pathophysiology [38, 41, 54–56]. This model requires intact human eyes for studying physiological parameters such as IOP homeostasis and aqueous humor dynamics. However, the high processing cost, limited availability of intact human eyes and higher failure rate has restricted the wider use of this model in glaucoma research. In this study, we developed an *ex-vivo* cultured human corneoscleral segment model to study glaucomatous TM damage. The human corneoscleral segment tissue is a gift that is given post-mortem for corneal transplant. Once excised, the tissue is tested to determine eligibility for transplantation. Due to stringent eligibility criteria, the majority of corneoscleral segments are deemed ineligible for transplant. These ineligible tissues are typically discarded due to low demand for research. The processing fees associated with obtaining corneoscleral segments from the eye bank is considerably lower compared to obtaining whole globes. An excised human corneoscleral segment is comprised of cornea, sclera, and the TM rim. Ciliary body, iris, and lens are promptly removed post-excision. Corneoscleral segments without an intact TM rim were not used for this study.

To determine optimal culture conditions, corneoscleral segments were divided into quadrants and cultured under standard and low serum conditions for 7 days. The structural integrity of the TM was preserved in stationary culture. Histochemical analysis of the quadrants showed no difference in TM morphology between standard (10% FBS) and low serum (0.5% FBS) conditions (**Fig 2**). TUNEL staining revealed a significant increase in TM cell death at low serum compared to the standard serum conditions. Interestingly, the number of TUNEL-positive cells were higher in adjacent scleral tissue than that of the TM region in quadrants cultured at low serum (**Fig 2**). Unlike the avascular TM, the vascular sclera appears to be highly serum dependent. Serum starvation can lead to changes in scleral cells, which as a consequence may adversely affect adjacent TM cells. Therefore, we decided to use 10% serum media to avoid any indirect effects of serum starvation on the TM or the adjacent tissues. The role of adjacent tissues should also be taken into consideration when interpreting the effects of drugs on the TM of corneoscleral quadrants. Previous studies with the anterior segment perfusion

culture model have used serum-free media for perfusion [34, 38, 39, 41, 57]. Continuous perfusion of fresh media provides constant supply of nutrients to the TM cells eliminating the need for serum in perfusion tissue cultures. The serum dependency of the corneoscleral quadrants in our stationary culture may be due to the lack of continuous perfusion of fresh media. In our study, we observed that serum supplementation was beneficial for the overall health of the tissue. However, we used low serum conditions (0.5% FBS) while treating quadrants with TGFβ2. Considering that serum contains growth factors including TGFβ2, increasing serum concentration might saturate the effect of TGFβ2 on the TM of corneoscleral quadrants. In support of this, we observed that TGFβ2 had minimum effects on ECM proteins when tissues were cultured at high serum compared to low serum (2% and 5% FBS). It is important to note that the prolonged time in culture (4–7 days) before receipt of these corneas may negatively affect tissue viability and cause cell death. However, previous studies have shown that preculture of tissues in stationary media for 7 days prior to perfusion aids in tissue stabilization [39, 55].

Progressive loss of TM cells has been reported with ageing and this loss appears to be accelerated in POAG [44, 45]. Several factors associated with glaucomatous pathology may play a role in accelerating TM cell loss and reduced tissue cellularity. For example, Clark et al. (1995) and Fleenor et al. (2006) observed an increase in TM cell loss in *ex vivo* perfused human anterior segments with Dex and TGFβ2 treatment respectively [34, 35].

A hallmark feature for a glaucoma model is its ability to mimic the glaucomatous TM pathology in response to glaucoma causing factors. The cultured human corneoscleral quadrants were treated with 3 different glaucoma factors to determine their effect on TM pathology. As expected, Dex treatment of anterior segments resulted in an increased ECM deposition in the TM. Furthermore, Dex treatment also induced ER stress in the TM of cultured human corneoscleral quadrants. A 30% IOP responsive rate was reported in Dex treated *ex-vivo* perfused human anterior segments [34]. In the present study, we did not evaluate the response rate as corneoscleral segment culture model is unfeasible to IOP and outflow measurements. Instead, we observed Dex-induced ECM changes in the TM only in 50% of cultured corneoscleral segments. Treatment with TGFB2, a glaucoma associated growth factor, resulted in an increased ECM deposition in the TM of corneoscleral segments. Unlike Dex, TGFB2-induced ECM changes in the TM were observed in all cultured quadrants. WT and mutant myocilin were expressed in the TM via lentiviral transduction. Consistent with our previous findings [13, 53], expression of mutant myocilin led to intracellular accumulation, induction of ER stress and increased deposition of ECM proteins in the TM of corneoscleral segments.

We observed that glaucoma causing factors significantly increased chronic ER stress markers, CHOP and ATF4 but not GRP78 in the TM tissues of corneoscleral segments. Although GRP78 expression showed an increasing trend, the difference compared to control was not significant. GRP78 is a key activator of ER stress sensors (IRE1, PERK and ATF6) that activates the UPR pathway. GRP78 is normally bound to these ER stress sensors, thus keeping them inactivated. Under stress conditions or accumulation of misfolded proteins, GRP78 is released from these sensors, resulting into activation of ER stress sensors and the UPR pathway [58]. Since GRP78 is expressed abundantly in TM tissues, a small increase after treatment with glaucoma factors may be sufficient to activate UPR pathway. This is apparent from significantly increased ATF4 and CHOP, which are downstream of GRP78 and are considered classical markers of chronic ER stress.

Although the human *ex-vivo* corneoscleral segment model is a promising prospect in modeling human glaucoma condition, there are certain limitations to be considered. Since the primary use of these corneoscleral segments is for human transplant, the portion of sclera posterior to the limbus is removed during the excision process. In addition to sclera, ciliary body,

iris, and lens are also removed. The removal of scleral tissue reduces the overall diameter of the corneoscleral segments, rendering them unusable for perfusion studies involving aqueous humor dynamics and IOP homeostasis. Furthermore, the removal of ciliary body may affect the TM contractile tone and can lead to biochemical changes in the TM cells. Of note, the ciliary body is also removed even in an *ex-vivo* human anterior segment perfusion culture to avoid the blockage of outflow with pigment originating from the degenerating pigmented cells from the ciliary body [57]. Unlike intact human eyes used for the perfusion culture (which can be delivered within 24-48hrs of the donor death) the human corneoscleral segments were delivered between 5–7 days. Despite this, we did not observe any notable changes in the TM morphology in these corneoscleral segments. The high variability between different donors is a common issue in research involving post-mortem human tissues. The corneoscleral segments used in this model can be divided into quadrants to perform control and experimental treatments on the same eye, thereby decreasing variability associated with contralateral control and reducing the overall cost of research. However, it is important to note that TM cellularity may differ along the circumferential length of the TM within the same eye. Furthermore, it is now known that the AH outflow through the TM is segmental with some areas along the circumferential TM length experiencing higher flow rate compared to other areas [40, 59, 60]. Therefore, tissues from multiple donors must be tested in order to account for spatial differences in cellularity and segmental flow along the TM. Despite these limitations, the *ex-vivo* cultured human corneoscleral explant model efficiently simulates glaucomatous pathology, and perhaps provides an attractive cost-efficient model for initial screening to study effects on the TM.

A similar *ex-vivo* culture method has been reported previously which utilized TM tissues recovered from discarded corneal rims after surgical corneal transplantation [61]. We have refined culture conditions over this previous study. Our current model differs from the previous study in several aspects: 1) We validate TM tissue integrity and cellularity of the cultured corneoscleral segment using H&E staining and TUNEL assays respectively. 2) We show the importance of serum to maintain the tissue integrity of corneoscleral segments in stationary culture. 3) In our study, we treated corneoscleral tissues with biologically relevant doses of Dex (100 nM) and TGFβ2 (5ng/ml) to study their effects on TM tissue in *ex vivo* cultured human corneoscleral segments while Gonzalez et al, 2012 treated TM rings with higher doses of Dex (250nM) and TGFβ1 (50nM). Of note, increased levels of TGFβ2 but not TGFβ1 in the aqueous humor are associated with ECM remodeling in the TM [14, 62–64]. 4) In addition to Dex and TGFβ2, we have shown that effects of mutant myocilin can be studied in intact TM tissues using this model. Considering limited availability of MYOC-associated POAG donor tissues, this approach offers the possibility of understanding the pathology behind MYOC-associated glaucoma. 5) Finally, we also examined the ER stress markers induced by the glaucoma factors.

## Supporting information

**S1 Data.**
(PDF)

## Acknowledgments

The authors thank the Lions Eye Institute (Tampa, Florida) for supplying the corneoscleral segments, and Sandra Neubauer for expert technical assistance. These studies were supported by the National Institutes of Health; EY028616 (GSZ) and EY026177 (GSZ).

## Author Contributions

**Conceptualization:** Pinkal D. Patel, Gulab S. Zode.

**Data curation:** Pinkal D. Patel.

**Formal analysis:** Ramesh B. Kasetti, Pinkal D. Patel, Gulab S. Zode.

**Funding acquisition:** Gulab S. Zode.

**Investigation:** Ramesh B. Kasetti, Prabhavathi Maddineni.

**Methodology:** Ramesh B. Kasetti, Pinkal D. Patel.

**Supervision:** Gulab S. Zode.

**Validation:** Ramesh B. Kasetti, Prabhavathi Maddineni.

**Writing – original draft:** Ramesh B. Kasetti, Pinkal D. Patel, Gulab S. Zode.

**Writing – review & editing:** Gulab S. Zode.

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
