## [Decision Letter · Decision Letter 0]

27 Dec 2019

PONE-D-19-32303

Ex-vivo cultured human corneoscleral segment model to study the effects of glaucoma factors on trabecular meshwork

PLOS ONE

Dear Dr. Zode,

Thank you for submitting your manuscript to PLOS ONE. After careful consideration, we feel that it has merit but does not fully meet PLOS ONE’s publication criteria as it currently stands. Therefore, we invite you to submit a revised version of the manuscript that addresses the points raised during the review process.

PLOS ONE now requires that submissions reporting blots or gels include original, uncropped blot/gel image data as a supplement or in a public repository. These requirements apply both to the main figures and to cropped blot/gel images included in Supporting Information. Please provide any missing raw image data for blot/gel results when submiting your revision. 

We would appreciate receiving your revised manuscript by Feb 10 2020 11:59PM. To enhance the reproducibility of your results, we recommend that if applicable you deposit your laboratory protocols in protocols.io, where a protocol can be assigned its own identifier (DOI) such that it can be cited independently in the future. For instructions see: http://journals.plos.org/plosone/s/submission-guidelines#loc-laboratory-protocols

We look forward to receiving your revised manuscript.

Kind regards,

Paloma B. Liton, PhD

Academic Editor

PLOS ONE

Journal Requirements:

2. We note that your study involved donated tissue/organs. Please provide the following information regarding tissue/organ donors for cases analyzed in your study.

1. Please provide the source(s) of the transplanted tissue/organs used in the study, including the institution name and a non-identifying description of the donor(s).

2. Please state in your response letter and ethics statement whether the transplant cases for this study involved any vulnerable populations; for example, tissue/organs from prisoners, subjects with reduced mental capacity due to illness or age, or minors.

- If a vulnerable population was used, please describe the population, justify the decision to use tissue/organ donations from this group, and clearly describe what measures were taken in the informed consent procedure to assure protection of the vulnerable group and avoid coercion.

- If a vulnerable population was not used, please state in your ethics statement, “None of the transplant donors was from a vulnerable population and all donors or next of kin provided written informed consent that was freely given.”

3. In the Methods, please provide detailed information about the procedure by which informed consent was obtained from organ/tissue donors or their next of kin. In addition, please provide a blank example of the form used to obtain consent from donors, and an English translation if the original is in a different language.

4. Please state whether the donors specifically provided consent for the use of their donated tissue for both transplantation and donation purposes.

5. Please indicate whether the donors were previously registered as organ donors. If tissues/organs were obtained from deceased donors or cadavers, please provide details as to the donors’ cause(s) of death.

6. Please provide the participant recruitment dates and the period during which transplant procedures were done (as month and year).

7. Please discuss whether medical costs were covered or other cash payments were provided to the family of the donor. If so, please specify the value of this support (in local currency and equivalent to U.S. dollars).

8. Please state whether you had access to any identifying information about the donors (e.g. names/addresses) as part of this study."

Reviewers' comments:

Reviewer's Responses to Questions

**Comments to the Author**

1. Is the manuscript technically sound, and do the data support the conclusions?

Reviewer #1: Partly

Reviewer #2: Partly

2. Has the statistical analysis been performed appropriately and rigorously? 

Reviewer #1: I Don't Know

Reviewer #2: No

3. Have the authors made all data underlying the findings in their manuscript fully available?

Reviewer #1: Yes

Reviewer #2: Yes

4. Is the manuscript presented in an intelligible fashion and written in standard English?

Reviewer #1: Yes

Reviewer #2: Yes

5. Review Comments to the Author

Reviewer #1: This interesting study discusses the potential of using ineligible human donor corneoscleral segments to model glaucoma. The authors used different factors known to be involved in glaucoma pathogenesis, or known to induce glaucoma and examined the TM tissue. Overall, this study shows a promising model for examining the trabecular meshwork tissue in ex vivo culture.

Comments:

1. In the introduction it states that current glaucoma treatment do not address the pathology of glaucomatous TM damage – Rho Kinase inhibitors are now in clinical use and have been shown to address this.

2. In the methods, a section regarding the statistical analyses conducted needs to be included.

3. There are two copies of figure 2 included – I’m not sure which one is to be published but in the second one, the tissue orientation needs to be the same as the H&E.

4. For the TUNEL assay analysis, I would like to see cell counts conducted by a masked observer as it appears to me that there are significantly more TUNEL positive cells in the TM tissues at 0.5% FBS compared to the 10% FBS. (also, the authors showing delineation of what they are determining is the TM might help readers)

5. Fig 3. Panel labeling is a bit confusing with 2 sets of A and B labels. I think just label the individual panels and not the set of panels. (i.e. take out the labels in black font.)

6. Not sure why there is immunostaining of ATF4 but no Western blot, and Western blot of GRP78 but no immunostaining. For consistency, I think it would be better to see both methods used to assess one or both proteins.

7. Figure 5 shows immunostaining of αSMA but there is not mention of this figure panel/result in the TGFβ2 results section. There is also no mention of GRP78 and it is shown by Western blot.

8. In all figures, labelling of the TM, SC and other visible tissues would be helpful.

9. For all the figures, if the brightfield is to be included in the merge, I would like to see the separate brightfield and immunostaining along with the merged image. Further, I’m not sure why only some figures have the brightfield while others do not? All the images with immunostaining should include the separate brightfield and brightfield merged images.

10. In the discussion, it is mentioned that the serum dependency of the segments may be due to lack of continuous perfusion, did the authors try changing the media of their stationary culture more often to determine if this was the case (every 12 or 24 hours)?

11. Did you observe any morphological changes in the TM with Dex or TGFβ2 treatment? Beam thickening or decreased trabecular spaces as well as the increase in ECM proteins?

12. In the discussion it is mentioned that no changes in TM structural integrity were observed between corneoscleral segments and intact human eyes. Is this data shown?

13. Did the authors compare TM cellularity between the control vs. treated quadrants to determine that it is/is not a factor in these studies?

14. In the discussion, the authors state that compared to the Gonzalez et al. study, the cultured the tissue in its native tissue architecture with minimal surgical intervention. I’m not sure what is meant by this – the previous study followed a similar method of segmenting and culturing tissue with/without treatments. I’d like to see the authors expand on this point.

15. There are some grammatical and spelling errors throughout the paper that need to be corrected. E.g. second line of the abstract “a most common form…” should be either “the most common form…” OR “a common form…”. The third line of the introduction “Majority of POAG patients…” should be “The majority of POAG patients…”

Reviewer #2: Current studies using human anterior segment perfusion organ culture to identify effects of glaucoma factors and disease modifying drugs on aqueous humor dynamics requires freshly enucleated intact human eyes. These are both expensive and have limited availability. This study explores the feasibility of using human donor corneoscleral segments, which are much more widely available and less expensive, for better understanding morphological and biochemical changes associated with POAG. Corneoscleral segments were dissected and cultured prior to treatment with either Dex, TGFβ2, or lentivirus. Then protein expression levels were analyzed to determine whether the TM tissues showed an appropriate response which would indicate the usefulness of the model system as a pre-screening tool for glaucoma therapeutics. The authors show that treatment of the corneoscleral segments with glaucoma factors lead to glaucomatous TM changes, suggesting that this model system which closely resembles the in vivo state of the tissues may be a good system in which to test disease modifying agents.

Overall, this study is of interest and may provide a cost-effective alternative to the need for intact human donor eyes to test potential glaucoma therapeutics. In spite of this, there are several concerns that need to be addressed, including the need for more quantitative methods in assessing cell differences between treatments and tissues, as well as ensuring statistical significance of the data with the appropriate number of biological replicates. Specific comments are provided below:

1) It is not clear whether the authors did any sort of statistical analyses to determine the significance of their data in Figures 2 and 3. Please include cell counts (for example, TUNEL positive cells versus total) to quantitate differences you are observing by eye. This should be included with the appropriate statistical test to determine if this observed difference is statistically significant. This would strengthen the manuscript.

2) In Figures 2 and 3 please provide text and/or arrows to indicate regions of interest. This especially because panels D-E in Figure 2, and panels E-H in Figure 3, are dark and hard to see the blue and green colors.

3) Please include arrows and/or text to indicate anatomical features in Figures 4 and 5, as well as to indicate areas of interest – it is not clear where the TM and SC are in these images and the orientation of the images is different from figure to figure in this manuscript.

4) Did the authors look at any other ER stress markers by Western blot other than GRP78 (Figure 4), such as CHOP or ATF4? It is not clear why the authors chose to show immunostaining for ATF4 but not by western blot. This data would be strengthened and more consistent if other stress markers were shown to be increased in response to Dex treatment by both western blot and immunostaining methods.

5) Please define what lysis buffer was used to lyse the TM tissues prior to running on gels for Western blots. Also, much of the ECM from TM tissues is not soluble in most lysis buffers. It is not clear whether the insoluble material was separated out before loading the gels. Please provide additional details as to whether the material run on gels included insoluble and soluble materials as this may affect the levels of proteins seen by Western blot.

6) Related to the prior point, in Figures 4, 5, and 6 there is Western blot data from TM tissues. The authors state that Coomassie staining of the gels was performed to ensure equal protein loading; however, this is not an appropriate method to normalize samples, especially since the authors are claiming that various proteins are present at different levels. Typically a BCA assay is used to measure total protein in the samples and they are normalized to these concentrations. Please include a more quantitative measure of the samples prior to loading on gels for Western blots.

7) Also for these three figures (4, 5, and 6) the authors state that they had n = 3 for lysates and n = 8 for conditioned medium. Please clarify whether this was three biological replicates or experimental replicates, since this is important information considering biological variability. It is important to show that these experiments have been done in a scientifically rigorous manner using tissues from multiple biological donor eyes.

8) Did the authors observe differences between donor eyes in terms of the observed response to Dex or TGFβ? It is difficult to determine whether all of the tissues tested were from Dex responders and if not, how this would affect the results. Please include additional comments.

6. PLOS authors have the option to publish the peer review history of their article (what does this mean?). If published, this will include your full peer review and any attached files.

Reviewer #1: No

Reviewer #2: No

---

## [Author Response · Author response to Decision Letter 0]

12 Feb 2020

Authors Response Letter

We very much appreciate the enthusiasm for this manuscript and also thank reviewers for providing valuable insights and suggestions. We feel that we have addressed the reviewers’ concerns as detailed in the response to the reviewers below:

Reviewer #1

1. In the introduction it states that current glaucoma treatment do not address the pathology of glaucomatous TM damage – Rho Kinase inhibitors are now in clinical use and have been shown to address this.

Authors response: We agree with the reviewer’s comment. We have now modified these statements and included a sentence about the newest class of glaucoma treating drugs (Rho kinase inhibitors), which directly act on trabecular meshwork to increase aqueous humor outflow facility (page 3). 

2. In the methods, a section regarding the statistical analyses conducted needs to be included.

Authors response: We have now included a separate section for the statistical analysis in the methods (page 9). In addition, we have included statistical analyses of Western blots in all figures.

3. There are two copies of figure 2 included – I’m not sure which one is to be published but in the second one, the tissue orientation needs to be the same as the H&E.

Authors response: We thank the reviewer for pointing out this oversight. We have now removed one copy of Figure 2. We have also now kept the same tissue orientation for both TUNEL and H&E stained images.

4. For the TUNEL assay analysis, I would like to see cell counts conducted by a masked observer as it appears to me that there are significantly more TUNEL positive cells in the TM tissues at 0.5% FBS compared to the 10% FBS. (also, the authors showing delineation of what they are determining is the TM might help readers)

Authors response: We appreciate the reviewer’s suggestion. The number of TUNEL positive cells over the total number of cells (DAPI stained) in the TM region were counted in a masked manner and the data was incorporated in Figure 2 and discussed in the result section (page 11). A significant increase in TUNEL positive cells was observed in 0.5% FBS cultured group compared to zero-day group. Moreover, TUNEL positive cells were increased in 0.5% FBS cultured group compared to 10% FBS group, however this difference was not statistically significant. We have also marked TM region by arrows.

5. Fig 3. Panel labeling is a bit confusing with 2 sets of A and B labels. I think just label the individual panels and not the set of panels. (i.e. take out the labels in black font.)

Authors response: We have now labeled each panel in Figure 3 individually as suggested. 

6. Not sure why there is immunostaining of ATF4 but no Western blot, and Western blot of GRP78 but no immunostaining. For consistency, I think it would be better to see both methods used to assess one or both proteins.

Authors response: We appreciate the reviewer’s suggestion. We have now performed additional experiments to maintain consistency both in immunostaining and Western blot data. Immunostaining for FN, Col-IV, Grp78 and Western blot analysis for FN, ATF4, Grp78 and CHOP were uniformly conducted in all 3 groups. 

7. Figure 5 shows immunostaining of αSMA but there is not mention of this figure panel/result in the TGFβ2 results section. There is also no mention of GRP78, and it is shown by Western blot.

Authors response: We thank the reviewer for pointing out this oversight. Both αSMA and GRP78 in the Figure 5 are now included in the result section (page #14).

8. In all figures, labelling of the TM, SC and other visible tissues would be helpful.

Authors response: We thank the reviewer for this suggestion. TM, SC and sclera are now clearly labeled in all figures.

9. For all the figures, if the brightfield is to be included in the merge, I would like to see the separate brightfield and immunostaining along with the merged image. Further, I’m not sure why only some figures have the brightfield while others do not? All the images with immunostaining should include the separate brightfield and brightfield merged images.

Authors response: As suggested, we have now included a separate brightfield images for all figures. We did not include the ‘immunostaining merged with brightfield’ since merging the immunostained images with the brightfield obscured the quality of immunostaining.

10. In the discussion, it is mentioned that the serum dependency of the segments may be due to lack of continuous perfusion, did the authors try changing the media of their stationary culture more often to determine if this was the case (every 12 or 24 hours)?

Authors response: We have changed the conditioned medium every 48hrs in a 7-day stationary culture. In ex-vivo perfusion cultured system or in vivo, there is a continuous flow of the culture medium through the conventional outflow path. The similar conditions may not be possible in the stationary culture system and we believe that changing the culture media for every 24hrs may not overcome the problem. 

11. Did you observe any morphological changes in the TM with Dex or TGFβ2 treatment? Beam thickening or decreased trabecular spaces as well as the increase in ECM proteins?

Authors response: We have limited the scope of our current study to examine the TM morphology by H&E staining. An ultrastructural (electron microscopic) examination was not conducted to determine the beam thickening or intercellular space in the TM. Based on H&E staining, we did not observe any gross morphological changes in the TM of Dex or TGFβ2 treated tissue compared to their respective controls. Using, immunohistochemical analysis, we further demonstrate that there is increased fibronectin and collagen deposition in Dex or TGFβ2-treated TM tissues compared to vehicle-treated TM (Figures 4-5).

12. In the discussion it is mentioned that no changes in TM structural integrity were observed between corneoscleral segments and intact human eyes. Is this data shown?

Authors response: We appreciate the reviewer pointing this oversight. We did not conduct a side by side comparison between the intact human eyes and cultured corneoscleral segments. We stated these points in the discussion based on our previous observation of H & E stained intact human eye sections. We have now removed this sentence from the discussion. 

13. Did the authors compare TM cellularity between the control vs. treated quadrants to determine that it is/is not a factor in these studies?

Authors response: As suggested, we have now compared TM cellularity (DAPI counting) between each treatment and observed no significant difference in TM cellularity between control and treated quadrants. We have stated these findings in the results (page # 13).

14. In the discussion, the authors state that compared to the Gonzalez et al. study, the cultured the tissue in its native tissue architecture with minimal surgical intervention. I’m not sure what is meant by this – the previous study followed a similar method of segmenting and culturing tissue with/without treatments. I’d like to see the authors expand on this point.

Authors response: We have now clarified the differences between ours and a previous study by Gonzalez et al.

15. There are some grammatical and spelling errors throughout the paper that need to be corrected. E.g. second line of the abstract “a most common form…” should be either “the most common form…”OR “a common form…”. The third line of the introduction “Majority of POAG patients…” should be “The majority of POAG patients…”

Authors response: We appreciate the reviewer pointing out these oversights. We have now corrected these errors and also carefully proofread the manuscript for grammatical errors.

Reviewer #2: 

1. It is not clear whether the authors did any sort of statistical analyses to determine the significance of their data in Figures 2 and 3. Please include cell counts (for example, TUNEL positive cells versus total) to quantitate differences, you are observing by eye. This should be included with the appropriate statistical test to determine if this observed difference is statistically significant. This would strengthen the manuscript.

Authors response: We appreciate the reviewer’s suggestion. We have now counted total number of TUNEL positive cells in the TM region and data is graphically shown in Figures 2G and 3I with appropriate statistical analyses. In addition, we have included a separate section in the Methods describing these statistical analyses.

2. In Figures 2 and 3 please provide text and/or arrows to indicate regions of interest. This especially because panels D-E in Figure 2, and panels E-H in Figure 3, are dark and hard to see the blue and green colors.

Authors response: As suggested, we have now clearly marked the regions of interests including TM and SC in all figures.

3. Please include arrows and/or text to indicate anatomical features in Figures 4 and 5, as well as to indicate areas of interest – it is not clear where the TM and SC are in these images and the orientation of the images is different from figure to figure in this manuscript.

Authors response: The regions of interest are now clearly marked, and the orientation of images was corrected in all figures.

4. Did the authors look at any other ER stress markers by Western blot other than GRP78 (Figure 4), such as CHOP or ATF4? It is not clear why the authors chose to show immunostaining for ATF4 but not by western blot. This data would be strengthened and more consistent if other stress markers were shown to be increased in response to Dex treatment by both western blot and immunostaining methods.

Authors response: We agree with the reviewer’s suggestion. We have now conducted additional experiments and included Western blot data for GRP78, ATF4 and CHOP for all 3 treatments (Dex/TGFβ2/MYOC). We have also included GRP78 immunostaining data in addition to FN and Col-IV in all the figures. To maintain the consistency throughout the manuscript, we have now removed the ATF4 immunostaining data from the Figure 4.

5. Please define what lysis buffer was used to lyse the TM tissues prior to running on gels for Western blots. Also, much of the ECM from TM tissues is not soluble in most lysis buffers. It is not clear whether the insoluble material was separated out before loading the gels. Please provide additional details as to whether the material run on gels included insoluble and soluble materials as this may affect the levels of proteins seen by Western blot.

Authors response: We have used commercial RIPA lysis buffer containing protease inhibitors to lyse the TM tissues. The insoluble material was separated out by centrifugation and supernatants were subjected to SDS-PAGE and Western blot analysis. Since RIPA buffer contains 0.1% SDS, it is likely that we are able to solubilize most of intracellular and extracellular ECM proteins. We have now included these details in the Methods (Page #8).

6. Related to the prior point, in Figures 4, 5, and 6 there is Western blot data from TM tissues. The authors state that Coomassie staining of the gels was performed to ensure equal protein loading; however, this is not an appropriate method to normalize samples, especially since the authors are claiming that various proteins are present at different levels. Typically a BCA assay is used to measure total protein in the samples and they are normalized to these concentrations. Please include a more quantitative measure of the samples prior to loading on gels for Western blots.

Authors response: We have utilized a Coomassie staining of the gels to examine the equal protein loading in the conditioned medium. We choose this because the phenol red and serum present in the media interferes with the accuracy of BCA assay. It should be noted that a similar approach has been utilized by several previous studies to ensure equal protein loading of the conditioned medium (Wordinger et al., 2007; Kasetti et al, 2016 and 2018; Jain et al., 2017).

7. Also for these three figures (4, 5, and 6) the authors state that they had n = 3 for lysates and n = 8 for conditioned medium. Please clarify whether this was three biological replicates or experimental replicates, since this is important information considering biological variability. It is important to show that these experiments have been done in a scientifically rigorous manner using tissues from multiple biological donor eyes.

Authors response: Experiments in Figures 4, 5 and 6 were conducted in TM tissues of 3 different donors (n=3 biological replicates). Similarly, the conditioned medium was also obtained from 8 different biological donor eyes (3 from tissues used to isolate TM tissue lysates and 5 from tissues that were fixed and used for immunostaining). We have now included this information clearly under each figure in the figure legends.

8. Did the authors observe differences between donor eyes in terms of the observed response to Dex or TGFβ? It is difficult to determine whether all of the tissues tested were from Dex responders and if not, how this would affect the results. Please include additional comments.

Authors response: In the present study, we observed differences between donor eyes in terms of Dex response. Dex treatment induced ECM changes in ~50% of the treated corneoscleral segments. In contrast, TGFB2 mediated induction of ECM was observed in all treated corneoscleral segments. Ideally, it would be better to evaluate Dex response in terms of IOP elevation. However, these corneoscleral segments are unfeasible for use in IOP or outflow measurements. We have included these points in the Discussion (page # 19). 

Editorial comments:

We note that your study involved donated tissue/organs. Please provide the following information regarding tissue/organ donors for cases analyzed in your study.

1. Please provide the source(s) of the transplanted tissue/organs used in the study, including the institution name and a non-identifying description of the donor(s).

Authors Response: We have procured the postmortem human donor corneas from the Lions Eye Institute for Transplant and Research, Tampa. The table below describes non-identifying donor information. 

Tissue ID Donor Age Male/Female OD/OS Race

8762 59 M OS White

8763 59 M OD White

9436 67 F OS White

9437 67 F OD White

9450 57 M OS Black

9451 57 M OD Black

0658 76 M OD White

0659 76 M OS White

0242 65 M OD White

0243 65 M OS White

0244 62 F OD White

0245 62 F OS White

0286 69 M OD White

0287 69 M OS White

0179 76 M OD Caucasian

0179 76 M OS Caucasian

0252 54 M OD Caucasian

0252 54 M OS Caucasian

0262 69 F OD Caucasian

0262 69 F OS Caucasian

5017 41 M OS Caucasian

5017 41 M OD Caucasian

4983 29 M OS Caucasian

4983 29 M OD Caucasian

12252 73 M OD Other

12252 73 M OS Other

11848 59 NP OD Caucasian

11848 59 NP OS Caucasian

17292 68 M OD Caucasian

17292 68 M OS Caucasian

17331 59 M OD Caucasian

17331 59 M OS Caucasian

0508 60 F OD Hispanic

0508 60 F OS Hispanic

0612 68 F OD Caucasian

0612 68 F OS Caucasian

2. Please state in your response letter and ethics statement whether the transplant cases for this study involved any vulnerable populations; for example, tissue/organs from prisoners, subjects with reduced mental capacity due to illness or age, or minors.

- If a vulnerable population was used, please describe the population, justify the decision to use tissue/organ donations from this group, and clearly describe what measures were taken in the informed consent procedure to assure protection of the vulnerable group and avoid coercion. If a vulnerable population was not used, please state in your ethics statement, “None of the transplant donors was from a vulnerable population and all donors or next of kin provided written informed consent that was freely given.”

Authors response: None of the transplant donors was from a vulnerable population and all donors or next of kin provided written informed consent that was freely given.

3. In the Methods, please provide detailed information about the procedure by which informed consent was obtained from organ/tissue donors or their next of kin. In addition, please provide a blank example of the form used to obtain consent from donors, and an English translation if the original is in a different language.

Authors response: The Lions Eye Institute for Transplant and Research (LEITR) is registered under the Eye Bank Association of America (EBAA), and it follows EBAA regulations to obtain donor tissues. We directly procure the deidentified postmortem tissues from LEITR, which obtains the required informed consent. We have contacted LEITR to provide further information regarding the informed consent obtained and tissue procurement procedure. 

4. Please state whether the donors specifically provided consent for the use of their donated tissue for both transplantation and donation purposes.

Authors response: Consent is obtained from donors by Lions Eye Bank.

5. Please indicate whether the donors were previously registered as organ donors. If tissues/organs were obtained from deceased donors or cadavers, please provide details as to the donors’ cause(s) of death.

Authors response: All tissues were obtained from organ donors.

6. Please provide the participant recruitment dates and the period during which transplant procedures were done (as month and year).

Authors response: Not applicable.

7. Please discuss whether medical costs were covered or other cash payments were provided to the family of the donor. If so, please specify the value of this support (in local currency and equivalent to U.S. dollars).

Authors response: Not applicable.

8. Please state whether you had access to any identifying information about the donors (e.g. names/addresses) as part of this study."

Authors response: We do not have access to identifying donor information. We have received only the non-identifying donor information sheets along with rejected donor corneas. 

Authors response: We have provided original uncropped and unadjusted images of all blots and gels reported in the manuscript figures as per the journal guidelines. 

Authors response: We have removed the sentence “data not shown” and also the phrase that refers to this data.

---

## [Decision Letter · Decision Letter 1]

27 Feb 2020

PONE-D-19-32303R1

Ex-vivo cultured human corneoscleral segment model to study the effects of glaucoma factors on trabecular meshwork

PLOS ONE

Dear Dr. Zode,

Thank you for submitting your manuscript to PLOS ONE. After careful consideration, we feel that it has merit but does not fully meet PLOS ONE’s publication criteria as it currently stands. Therefore, we invite you to submit a revised version of the manuscript that addresses the points raised during the review process.

We would appreciate receiving your revised manuscript by Apr 12 2020 11:59PM. To enhance the reproducibility of your results, we recommend that if applicable you deposit your laboratory protocols in protocols.io, where a protocol can be assigned its own identifier (DOI) such that it can be cited independently in the future. For instructions see: http://journals.plos.org/plosone/s/submission-guidelines#loc-laboratory-protocols

We look forward to receiving your revised manuscript.

Kind regards,

Paloma B. Liton, PhD

Academic Editor

PLOS ONE

Reviewers' comments:

Reviewer's Responses to Questions

**Comments to the Author**

1. If the authors have adequately addressed your comments raised in a previous round of review and you feel that this manuscript is now acceptable for publication, you may indicate that here to bypass the “Comments to the Author” section, enter your conflict of interest statement in the “Confidential to Editor” section, and submit your "Accept" recommendation.

Reviewer #1: (No Response)

Reviewer #2: All comments have been addressed

2. Is the manuscript technically sound, and do the data support the conclusions?

Reviewer #1: Partly

Reviewer #2: Yes

3. Has the statistical analysis been performed appropriately and rigorously? 

Reviewer #1: Yes

Reviewer #2: Yes

4. Have the authors made all data underlying the findings in their manuscript fully available?

Reviewer #1: Yes

Reviewer #2: Yes

5. Is the manuscript presented in an intelligible fashion and written in standard English?

Reviewer #1: Yes

Reviewer #2: Yes

6. Review Comments to the Author

Reviewer #1: The authors have made significant improvements to the study however, in its current form, the data does not support the conclusions. There are some details that need to addressed further.

1. GRP78 is not significant under any of the treatments in the study as measured by Western blot. This needs to be discussed or more experiments need to be conducted to investigate this.

2. Under Dex and mutant MYOC treatments, cellular fibronectin is not significant, even though the secreted form is. This needs to be addressed in the discussion, or more experiments conducted to investigate.

3. Could the authors discuss why they investigated secreted ColIV in conditioned media and not the cellular form by Western blot? I’d like to see cellular ColIV measured by Western blot for comparison.

4. In Fig 6B. there is no visible MYOC present in the indicated TM region under the lentiviral treatment. This does not support the author’s statements regarding accumulated cellular mutant MYOC in TM cells.

5. Finally, the sentence regarding comparison of the TM structural intergrity of these corneoscleral quadrants to intact human eyes is still included the discussion

Reviewer #2: (No Response)

7. PLOS authors have the option to publish the peer review history of their article (what does this mean?). If published, this will include your full peer review and any attached files.

Reviewer #1: No

Reviewer #2: No

---

## [Author Response · Author response to Decision Letter 1]

3 Apr 2020

Authors Response Letter

We appreciate the reviewers’ valuable insights and suggestions which have substantially improved our manuscript. We have carefully addressed all concerns and performed additional experiments to address these concerns. These changes are highlighted in blue in the revised manuscript. A detailed response to the reviewer’s concerns is provided below:

1. GRP78 is not significant under any of the treatments in the study as measured by Western blot. This needs to be discussed or more experiments need to be conducted to investigate this.

Authors response: We agree with the reviewer’s comment and are thankful for their suggestion. Although there is a trend showing an increase in GRP78 expression in response to treatment with glaucoma factors, this increase is not significant in Western blot analysis. Please note that immunostaining for GRP78 as shown in Figures 4B, 5B and 6C clearly demonstrate that glaucoma factors increase GRP78 staining in the TM. We have added this clarification in the results and further included the following explanation in the discussion: 

“We observed that glaucoma causing factors significantly increased chronic ER stress markers, CHOP and ATF4 but not GRP78 in the TM tissues of corneoscleral segments. Although GRP78 expression showed an increasing trend, the difference compared to control was not significant. GRP78 is a key activator of ER stress sensors (IRE1, PERK and ATF6) that activates the UPR pathway. GRP78 is normally bound to these ER stress sensors, thus keeping them inactivated. Under stress conditions or accumulation of misfolded proteins, GRP78 is released from these sensors, resulting into activation of ER stress sensors and UPR pathway [1]. Since GRP78 is expressed abundantly in TM tissues, a small increase after treatment with glaucoma factors may be sufficient to activate the UPR pathway. This is apparent from significantly increased ATF4 and CHOP, which are downstream of GRP78 and are considered classical markers of chronic ER stress.” 

2. Under Dex and mutant MYOC treatments, cellular fibronectin is not significant, even though the secreted form is. This needs to be addressed in the discussion, or more experiments conducted to investigate.

Authors response: We thank the reviewer for their suggestion. In the present study, we observed differences in Dex response between donor eyes. Dex treatment-induced ECM changes were observed in ~50% of the treated corneoscleral segments. In our previous version of this manuscript, we included data from the non-responders as well as Dex responders for densitometric analysis. We re-analyzed the data using only the Dex responders and observed a significant increase in FN expression in the lysates (Figure 4D). For the mutant MYOC treatments, 3 additional biological replicates were run and analyzed to show significant change in FN expression (Figure 6E). We have edited the results and figures to include this new data.

3. Could the authors discuss why they investigated secreted ColIV in conditioned media and not the cellular form by Western blot? I’d like to see cellular ColIV measured by Western blot for comparison.

Authors response: We agree with the reviewer’s comment and appreciate their constructive feedback. The TM tissue peeled from each quadrant is only sufficient to run a single Western blot. Furthermore, the ColIV antibody did not work on the stripped blots that were previously probed with FN antibody since the band sizes for both FN and ColIV fall in the same vicinity. This is why we were only able to probe for FN in lysates, whereas FN and ColIV both were analyzed in the conditioned medium, which was available in sufficient quantities for multiple Western blots.

4. In Fig 6B. there is no visible MYOC present in the indicated TM region under the lentiviral treatment. This does not support the author’s statements regarding accumulated cellular mutant MYOC in TM cells.

Authors response: We thank the reviewer for their comment. We have now revised Figure 6B with new images, which clearly show increased mutant myocilin accumulation compared to WT MYOC. 

5. Finally, the sentence regarding comparison of the TM structural integrity of these corneoscleral quadrants to intact human eyes is still included the discussion

Authors response: We sincerely regret this oversight and as per the reviewer’s suggestion, we have made the correction by removing the statement from the discussion (Page 21, line 5) and replaced with the following statement. Note that we mentioned TM morphology since we examined TM morphology via H&E staining.

“Despite this, we did not observe any notable changes in the TM morphology in these corneoscleral segments.”

References:

1. Roussel BD, Kruppa AJ, Miranda E, Crowther DC, Lomas DA, Marciniak SJ. Endoplasmic reticulum dysfunction in neurological disease. Lancet Neurol. 2013;12(1):105-18. Epub 2012/12/15. doi: 10.1016/S1474-4422(12)70238-7. PubMed PMID: 23237905.

---

## [Editor Report · Decision Letter 2]

8 Apr 2020

Ex-vivo cultured human corneoscleral segment model to study the effects of glaucoma factors on trabecular meshwork

PONE-D-19-32303R2

Dear Dr. Zode,

We are pleased to inform you that your manuscript has been judged scientifically suitable for publication and will be formally accepted for publication once it complies with all outstanding technical requirements.

With kind regards,

Paloma B. Liton, PhD

Academic Editor

PLOS ONE
---

## [Editor Report · Acceptance letter]

13 Apr 2020

PONE-D-19-32303R2 

Ex-vivo cultured human corneoscleral segment model to study the effects of glaucoma factors on trabecular meshwork 

Dear Dr. Zode:

I am pleased to inform you that your manuscript has been deemed suitable for publication in PLOS ONE. Congratulations! Your manuscript is now with our production department. 

With kind regards,

on behalf of

Dr. Paloma B. Liton 

Academic Editor

PLOS ONE